# [Proposal-ML]
# Mimicking Humanity: A synthetic data-based approach to voice cloning in Text to Speech Systems

**Michael Wang**
DCST
Tsinghua University
2024280208

**Algazinov Aleksandr**
DCST
Tsinghua University
2024280037

**Joydeep Chandra**
DCST
Tsinghua University
2024280035

## 1 Background

While text-to-speech (TTS) technology has greatly improved in recent years, traditional TTS models often rely on large datasets of recorded human speech to produce accurate and natural synthetic speech. However, obtaining high-quality data at scale presents challenges such as high costs, user privacy concerns, lack of linguistic diversity, and data paucity.

Speech data remains inaccessible for many languages and dialects, leading to disparities in TTS quality across languages and accents.

This project aims to train a voice cloning model for use with TTS systems primarily using synthetic training data. The objective is to create a voice cloning model that can generalize well without reliance on human-produced training data.

The model is intended to be small enough to permit local inference. This will allow users to generate speech with their voice without granting legal rights to their data to a third party, as is currently the case with commercially available voice generation platforms. Furthermore, if generalizable, this approach of using synthetic training data can reduce the cost of acquiring training data when applied to the training of other categories of ML models.

## 2 Related Work

Several important TTS models include WaveNet [12], Tacotron [22], and FastSpeech [2]. WaveNet is an autoregressive model capable of producing high-quality, natural-sounding speech. However, since it is an autoregressive model, it is computationally inefficient. Tacotron is a seq2seq model with attention, making it faster compared to WaveNet. However, the model still faces issues with robustness and controllability, and it is still computationally demanding. The FastSpeech model shows better performance compared to Tacotron 2 (in terms of the listed metrics), while being faster.

While all of these models are capable of generating high-quality speech, they are limited to voices associated with pre-trained speakers. Zero-shot TTS systems aim to add the ability to generate speech for unknown speakers with only a short voice sample.

VALL-E [18] and VALL-E 2 [19] are zero-shot synthesis models that include direct TTS capability. Unfortunately, the source code associated with these models is not publicly available. OpenVoice

37th Conference on Neural Information Processing Systems (NeurIPS 2023).

[20], which we take inspiration from for our system architecture, is a newer model that aims to handle the voice cloning aspect without needing to directly handle any TTS function.

TTS models typically use datasets derived from LibreVox, which is a public domain audiobook repository, for training. However, synthetic data [21] has been successfully used for training speech recognition models, which suggests that synthetic data can be used as part of training other types of models as well.

## 3  Proposed Method

The primary focus is on developing a model that can take any speech audio and modify it so that the modified audio sounds as if it was spoken by some target speaker.

We intend for the model to have two inputs. The first is the audio to be transformed. Typically this will be audio generated by some TTS model, but human-spoken audio should also be a valid input. The second is a voice sample of the target speaker, which is used to extract unique characteristics of the target speaker's voice, such as pitch, tone, and timbre. Ideally, this should not need to be longer than a few seconds.

To clone the target speaker's voice, the model generates an embedding that represents the features extracted from the voice sample. Then, the input audio is passed through a voice conversion block, which modifies the input audio to match the speaker's embedding characteristics.

The final output is speech audio that sounds like it was spoken by the target speaker, retaining the linguistic content of the original synthesized audio but transformed to the voice characteristics of the provided speaker sample.

Synthetic data will be used to pre-train the model, reducing the need for extensive real-world recordings. By pre-training with a broad range of synthetic voices, we expect our model to generalize better to a variety of speaker profiles with minimal fine-tuning.

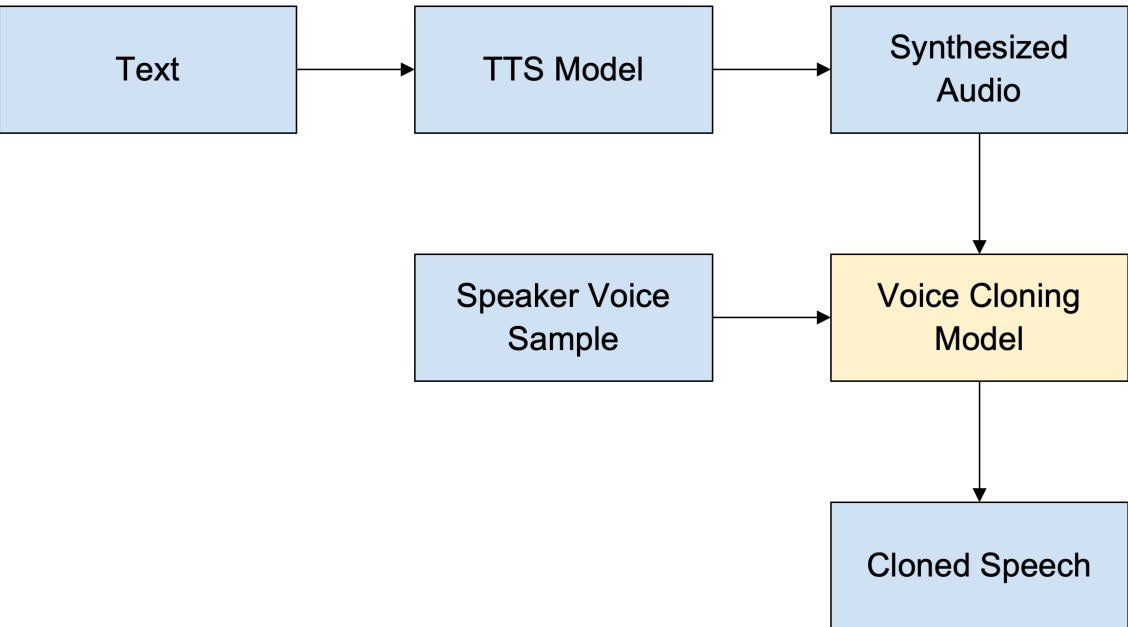

Figure 1: Architecture of a voice cloning system. The system this project is focused on is marked in yellow.

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
