# OpenReview forum: "[Proposal-ML] Mimicking Humanity: A synthetic data-based approach to voice cloning in Text to Speech Systems"
_tsinghua.edu.cn/THU/2024/Fall/AML — THU 2024 Fall AML Submission_

### Official Review · ~Thomas_Adler2 · 2024-11-06
**Straight to the point**

**Rating:** 10
**Confidence:** 4

**Review:**

Proposal is easy to read, clear and straight to the point. The methodology is well explained and we can see how this approach is different compared to previous methods (using synthetic data makes it more generalisable, aim of making it more computationally efficient etc)

---

### Official Review · ~Chua_Shei_Pern1 · 2024-11-06
**Clear approach**

**Rating:** 10
**Confidence:** 4

**Review:**

The proposal had a very strong and solid proposed method, which is commendable. The background and methodology provides good context to understand the project.

---

### Official Review · ~Huajun_Bai1 · 2024-11-07
**Synthetic Data-Driven Voice Cloning in TTS: Privacy, Innovation, and Challenges**

**Rating:** 7
**Confidence:** 3

**Review:**

Strengths

1. Innovative Use of Synthetic Data: The proposal to train a voice cloning model for TTS systems using synthetic data is a forward-thinking approach that addresses the challenges of data acquisition, including cost, privacy, and linguistic diversity. This method has the potential to democratize access to high-quality TTS systems.

2. Privacy-Preserving Local Inference: The focus on creating a model that allows for local inference without the need to share user data with third parties is a significant strength. It addresses the growing concern over data privacy and provides a more secure alternative to current commercial voice generation platforms.

3. Generalizability and Cost Reduction: The proposal's aim to develop a model that can generalize well without reliance on human-produced training data is commendable. If successful, this approach could reduce the cost of acquiring training data for other ML models, making it a valuable contribution to the field of machine learning.

Weaknesses

1. Technical Details on Synthetic Data Generation: While the proposal highlights the use of synthetic data, it lacks specific details on how this synthetic data will be generated and validated for quality. Ensuring that synthetic data accurately mimics the characteristics of real human speech is crucial for the success of the model.

2. Evaluation Metrics and Benchmarks: The proposal does not discuss how the performance of the voice cloning model will be evaluated against existing TTS models or industry standards. Defining clear evaluation metrics and benchmarks is essential for demonstrating the effectiveness of the proposed system.

---

### Official Review · ~Anton_Johansson1 · 2024-11-08
**Well written proposal**

**Rating:** 9
**Confidence:** 4

**Review:**

Overall it is av very interesting topic that has the potential to be truly helpful if successful. You have an innovative use of synthetic data to reduce dependence on extensive human speech datasets. However, your proposal lacks some technical details on model architecture, particularly the voice conversion block and embeddings. To strengthen your proposal, it could include a detailed evaluation strategy, perhaps using metrics like mean opinion scores, and discuss challenges in synthetic data quality. I understand that the page limit make it challenging to include all these details, but these points could be useful to consider for the final project. Very good job overall.

---

### Official Review · ~Liutao7 · 2024-11-09
**A Potential Synthetic Data-Based Voice Cloning Method, but Some Technical Challenges Need to Be Overcome**

**Rating:** 9
**Confidence:** 4

**Review:**

The proposal presents a voice cloning method based on synthetic data, aiming to address the issue of traditional speech synthesis models' dependence on a large amount of real voice data, and to improve the generalization ability of voice cloning and user privacy protection. The proposal demonstrates high integrity and creativity, and proposes a clear model architecture and training plan.
Advantages:
Integrity and Creativity: The proposal clearly articulates the research background, related work, method design, and expected goals, and proposes an innovative solution using synthetic data for pre-training.
Workload: The proposal shows a deep understanding of related literature and proposes a feasible technical roadmap, indicating that the researchers have done thorough preparatory work.
Areas for Improvement:
Lacks description of some key technical details, such as the voice conversion block and embedding; an evaluation system should be added, selecting appropriate metrics to assess the voice quality, naturalness, and similarity to the target speaker.

---

### Official Review · ~Diego_Cerretti1 · 2024-11-10
**Clear, relevant and well-written**

**Rating:** 10
**Confidence:** 4

**Review:**

The authors propose a voice cloning model for text-to-speech systems that uses synthetic training data to enable personalized, privacy-focused speech synthesis. The proposed method is clearly motivated and aligns with the current trends in TTS research. The objective and the technical approach are well-defined.

---

### Official Review · ~Rim_El_Filali1 · 2024-11-11
**Promising Voice Cloning with Synthetic Data but Needs Clearer Evaluation Metrics**

**Rating:** 9
**Confidence:** 4

**Review:**

This proposal presents a voice cloning model designed to enable personalized TTS experiences while addressing the common challenges of high data acquisition costs, privacy concerns, and linguistic diversity in traditional TTS. The model uses synthetic training data to avoid reliance on human speech recordings, aiming for a generalized voice conversion model with minimal real-world data.

Pros:
- Tackling privacy in voice generation by allowing users to clone voices locally, without third-party data rights, is a timely and impactful goal.
- The goal of using short voice samples to perform voice cloning across a variety of speakers with minimal fine-tuning could make the model highly adaptable to new users.

Cons:
- Limited details are provided regarding the model’s computational requirements and how it will ensure high-quality voice cloning in real-time on local devices.
- The proposal lacks specifics on how the model’s success will be quantitatively evaluated. Performance metrics would be helpful to understand its potential effectiveness.

---

### Official Review · ~Gausse_Mael_DONGMO_KENFACK1 · 2024-11-11
**Ambitious goal, perfectible path**

**Rating:** 8
**Confidence:** 4

**Review:**

The paper proposes a method for voice cloning in TTS systems. The approach aims to reduce the need for extensive human speech datasets, thereby lowering costs and protecting user privacy.  The model would take an audio file and a brief voice sample to generate speech that mimics the target speaker’s voice.

strength: Since data is central to most current ML methods, a generalizable approach enabling the use of synthetic data would be highly beneficial for the field.

weakness: The paper would benefit from additional clarity on the model architecture and audio data representation.

---

### Official Review · ~Suraj_Joshi2 · 2024-11-12
**Review on Zero-Shot Locally Inferable Voice Cloning Model**

**Rating:** 9
**Confidence:** 4

**Review:**

This project proposal proposes a good solution for generating real voices without requiring the target user to give their voice samples to third party. To achieve this goal, they propose idea of model that can run locally on edge devices, which can reduce the processing costs significantly. This model can be real helpful for low resource languages, which don't have publicly available large scale datasets for training large models, that are the current State of The Art models. Also, training on synthetic data can reduce the cost of data collection.

Here are few things that would have made proposal a great one:
1. The proposal clearly demonstrates the inference pipeline, but I could not get how the model would be trained, what would be inputs to the model in the training phase.
2. How the model would compensate the effect of using synthetic dataset for training is also bit unclear from the proposal.
3. In the proposal authors mention, if training the model using synthetic data is successful in this case, the training methodology can be applied to other domains too. But I guess success of training model using synthetic data does not necessarily imply success on other domains too. There are multiple domains, where training models using synthetic data have been practiced for long time (like robotics) but these domains face a great issue to to Sim2Real gap.
4. Finally some information how to make model that can run inference on edge devices would also have made the proposal a great one.

---

### Official Review · ~Kittaphot_Saengprachathanarak1 · 2024-11-12
**Review of "Mimicking Humanity"**

**Rating:** 10
**Confidence:** 4

**Review:**

This proposal outlines a novel approach to voice cloning in text-to-speech (TTS) systems, focusing on using synthetic data to reduce reliance on large, human-generated speech datasets. The authors aim to develop a model that can take input audio and modify it to match the characteristics of a target speaker, using only a short voice sample. By leveraging synthetic data for pre-training, the model can generalize across various speaker profiles without extensive fine-tuning, thus addressing challenges related to data scarcity and privacy concerns. The approach is promising, particularly in its potential to create smaller, more efficient models for local inference. However, further validation through experiments and comparison with existing methods like OpenVoice and VALL-E would strengthen the proposal. Overall, the methodology is well-structured and offers significant contributions to voice cloning in TTS systems, with potential for broader applications in privacy-preserving speech synthesis.

---

### Official Review · ~Grace_Xin-Yue_Yi1 · 2024-11-12

**Rating:** 8
**Confidence:** 4

**Review:**

This proposal defines the key challenges of TTS models, outlining the benefits of using synthetic data to address issues like privacy and the high costs of traditional models. Related works analyze the limitations of existing TTS models such as WaveNet, Tacotron, FastSpeech, and zero-shot synthesis models like VALL-E and OpenVoice. The discussion is thorough, addressing each model's strengths and weaknesses, particularly around computational efficiency and generalization capabilities. The proposed methodology is innovative but could benefit from including more technical details such as the data generation process, how the methodology overcomes the limitations of existing models mentioned in the related work section, and specifying evaluation metrics.